# Cryo-EM structure in situ reveals a molecular switch that safeguards virus against genome loss

Oliver W Bayfield[1,2]*, Alasdair C Steven[2], Alfred A Antson[1]*

[1]York Structural Biology Laboratory, Department of Chemistry, University of York, York, United Kingdom; [2]Laboratory of Structural Biology Research, National Institute of Arthritis Musculoskeletal and Skin Diseases, National Institutes of Health, Bethesda, United States

**Abstract** The portal protein is a key component of many double-stranded DNA viruses, governing capsid assembly and genome packaging. Twelve subunits of the portal protein define a tunnel, through which DNA is translocated into the capsid. It is unknown how the portal protein functions as a gatekeeper, preventing DNA slippage, whilst allowing its passage into the capsid, and how these processes are controlled. A cryo-EM structure of the portal protein of thermostable virus P23-45, determined in situ in its procapsid-bound state, indicates a mechanism that naturally safeguards the virus against genome loss. This occurs via an inversion of the conformation of the loops that define the constriction in the central tunnel, accompanied by a hydrophilic–hydrophobic switch. The structure also shows how translocation of DNA into the capsid could be modulated by a changing mode of protein–protein interactions between portal and capsid, across a symmetry-mismatched interface.

*For correspondence:
oliver.bayfield@york.ac.uk (OWB);
fred.antson@york.ac.uk (AAA)

**Competing interests:** The authors declare that no competing interests exist.

## Introduction

Tailed bacteriophages constitute the majority of viruses in the biosphere (*Bergh et al., 1989*; *Michaud et al., 2018*) and are a significant component of the human microbiome (*Shkoporov and Hill, 2019*). During assembly, these viruses translocate their genomic double-stranded DNA through a portal protein that occupies a single vertex of an icosahedral capsid. A similar mechanism is employed by the evolutionarily related herpesviruses (*McElwee et al., 2018*; *Trus et al., 2004*). Structural information about the portal protein is important not only for deducing the mechanism of capsid assembly (*Chen et al., 2011*), but also for understanding molecular events associated with genome translocation into preformed capsids (*Mao et al., 2016*; *Sun et al., 2015*; *Sun et al., 2008*), and genome ejection during infection (*Wu et al., 2016*). Although structures of isolated portal proteins, without the native capsid environment, have been determined to near-atomic resolution by X-ray crystallography and cryo-electron microscopy (*Lebedev et al., 2007*; *Lokareddy et al., 2017*; *Simpson et al., 2000*; *Sun et al., 2015*), a number of observations concerning these structures have yet to be rationalised in the context of the portal's many functional roles, including: the variable diameter of the central tunnel and flexibility of tunnel loops (*Lebedev et al., 2007*; *Simpson et al., 2000*; *Sun et al., 2015*); the symmetry mismatch between the portal and capsid vertex (12-fold versus 5-fold) (*Simpson et al., 2000*; *Sun et al., 2008*); and the portal's role in DNA translocation (*Harvey, 2015*; *Ray et al., 2010*). The influence of the properties of the internal tunnel, and how these can be modulated by external factors to coordinate DNA translocation, remains unclear. Cryo-EM studies on mesophilic herpesviruses characterised the shape of the portal protein tunnel in the mature virion and showed how DNA can be locked inside (*Liu et al., 2019*; *McElwee et al., 2018*). However, there are no detailed structural data on portal proteins in situ for unexpanded capsids,

primed for DNA packaging. Moreover, it has proven difficult to derive accurate models for tunnel loops of the portal protein, such as in the case of tailed bacteriophage φ29, where the flexible nature of the tunnel loops prevented their observation in a crystal structure (*Simpson et al., 2000*) and also in cryo-EM structures of the procapsid and mature capsid (*Xu et al., 2019*).

To gain knowledge about the structure of the dynamic DNA tunnel, we utilised a thermostable bacteriophage, P23-45. Thermophilic viruses must package their genomes under extreme temperature, imposing additional challenges compared to their mesophilic counterparts. This *Thermus thermophilus* bacteriophage is one of the few viruses for which conditions for packaging DNA into capsids in vitro have been established, and where isolated empty capsids were demonstrated to be competent at packaging DNA (*Bayfield et al., 2019*). Previous cryo-EM reconstructions of procapsids (unexpanded) and mature capsids (expanded), in which icosahedral symmetry was imposed, have revealed the extent of conformational changes that the major capsid proteins undergo upon capsid maturation. During the transition, the capacity of the capsid nearly doubles (*Bayfield et al., 2019*). In this study, the structure of the portal protein in situ, and analysis of the reconstruction of the unexpanded procapsid without imposing icosahedral symmetry, reveal substantive conformational differences in the structure of the portal protein (*Bayfield et al., 2019*). The most remarkable difference, induced in situ, is an inversion in the conformation of tunnel loops of the portal protein. The tunnel loop inversion 'switches' the surface properties at the tunnel's constriction from hydrophobic to hydrophilic and creates a wider opening. These observations indicate that the capsid shell plays a role in defining the conformation and properties of the portal protein, modulating DNA translocation into capsid.

## Results

### Structure of the in situ procapsid portal

P23-45 procapsids were purified from lysates of infected *Thermus thermophilus* cells (*Figure 1A*). The procedures used for cryo-EM data collection and computing the icosahedrally averaged reconstruction were described earlier (*Bayfield et al., 2019*). The in situ structure of the portal protein within the procapsid was determined by localised reconstruction of portal-containing vertices to a resolution of 3.7 Å by averaging around the 12-fold symmetric axis (*Supplementary file 1*, *Figure 1—figure supplement 1*; *Ilca et al., 2015*). The portal protein oligomer is a ring of 12 subunits (*Figure 1B,C*, *Video 1*), with each subunit folded into Crown, Wing, Stem, Clip, and Tunnel loop domains (*Figure 1D*). Most amino acid side-chains were clearly resolved in the map (*Figure 1E*), enabling construction of an accurate atomic model (PDB 6QJT). Comparison with the crystal structure of the portal protein from the closely related phage G20c (PDB 6IBG, 99.3% sequence identity) reveals several significant structural differences: notably, in the positions of the Crown and Wing domains and in the conformation of the tunnel loops (*Figure 2*, *Video 2*). In the in situ structure, the C-terminal Crown domain (residues 377–436) is shifted upwards away from the main body by ~5 Å (*Figure 2A,B*), and twisted by ~13° around the central axis (*Figure 2C*, *Video 2*). The Wing domain pivots ~8° downwards, towards the Clip (*Figure 2B*, *Video 2*). Although the two portal proteins compared are from different phages, they have closely related sequences. The most conservative substitution, I328V, is located in the tunnel, and two additional conservative substitutions, S189N and S367G, are located at the outer surface of the Wing in residues with solvent-exposed side chains. Such mutations are unlikely to bring about the observed differences in conformation.

### Differences between the portal conformations in the in situ and crystal structures

The most pronounced conformational differences seen in the in situ structure are in the tunnel loops (*Figure 2*). The tunnel diameter at its most constricted part is wider by ~8 Å (*Figure 2E,F*). Hydrophobic residues V325 and I330 are no longer exposed to the tunnel as they are in the crystal structure and are replaced by polar residues Q326 and N329 due to inversion in the tunnel loop conformation (*Figure 2D*). Residues 330–335, which protrude into the tunnel and are part of the longest helix in the crystal structure, instead adopt an extended loop conformation in situ (*Figure 2D*), facilitating the tunnel loop remodelling. These modifications alter the shape and surface properties of the tunnel, which widens and changes from hydrophobic to hydrophilic (compare *Figure 2E,F*).

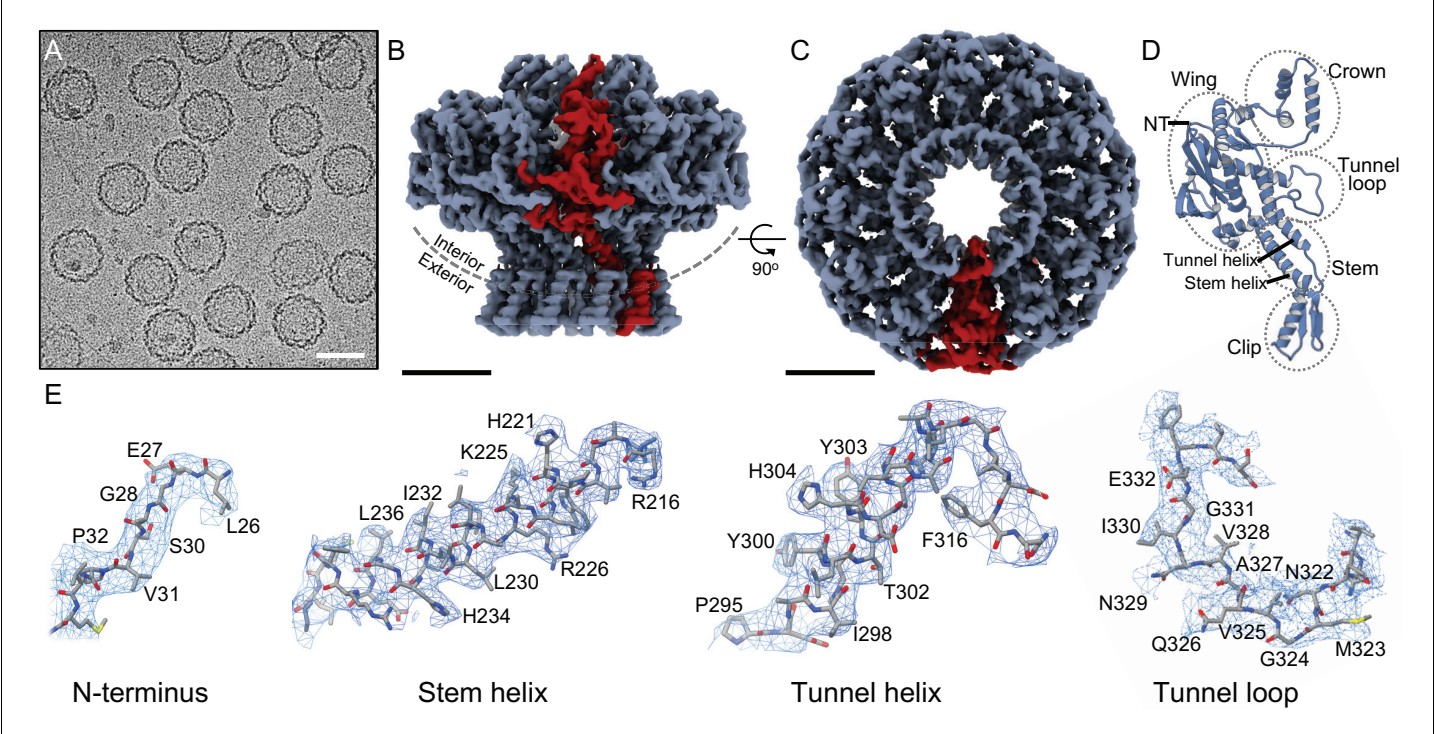

**Figure 1.** Structure of the portal protein in situ. (A) Cryo-electron micrograph of P23- 45 procapsids, scale bar 50 nm. (B) Cryo-EM reconstruction map with one subunit coloured red, scale bar 50 Å, and same for (C) but rotated 90°, viewed along 12-fold axis. (D) Ribbon diagram of one portal protein subunit. (E) Regions of the map and corresponding atomic models with residue numbers.

The online version of this article includes the following figure supplement(s) for figure 1:

**Figure supplement 1.** FSC curve for the portal protein reconstruction.

**Figure supplement 2.** Mass spectrometry analysis of the portal protein from unexpanded capsids.

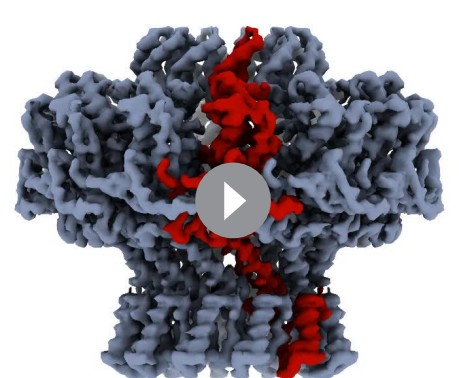

**Video 1.** Reconstruction of the in situ portal. Surface rendering, first viewed perpendicular to the tunnel axis, then viewed along the tunnel axis.

https://elifesciences.org/articles/55517#video1

The first N-terminal residue that could be reliably modelled in the in situ reconstruction was Leu26 (*Figure 2D*), in common with the crystal structure (PDB 6IBG). Mass spectrometry detected N-terminal residues of the portal protein subunits (*Figure 1—figure supplement 2*), indicating that the 25-amino acid N-terminal segment is present in at least some chains, but adopts variable conformations. Although the first residue with a defined conformation points towards the interior of the capsid in P23-45, it cannot be ruled out that the flexible N-terminal segment folds back and contributes to portal-capsid interactions.

## Portal–capsid interactions

The portal–capsid interface is spacious, with only relatively small surface areas of the portal's, within the Wing and Clip domains, engaged in interactions with the capsid (*Figure 3A,B*). Fitting of the C12-symmetrised portal reconstruction, presented here, into the asymmetric procapsid reconstruction (*Bayfield et al., 2019*),

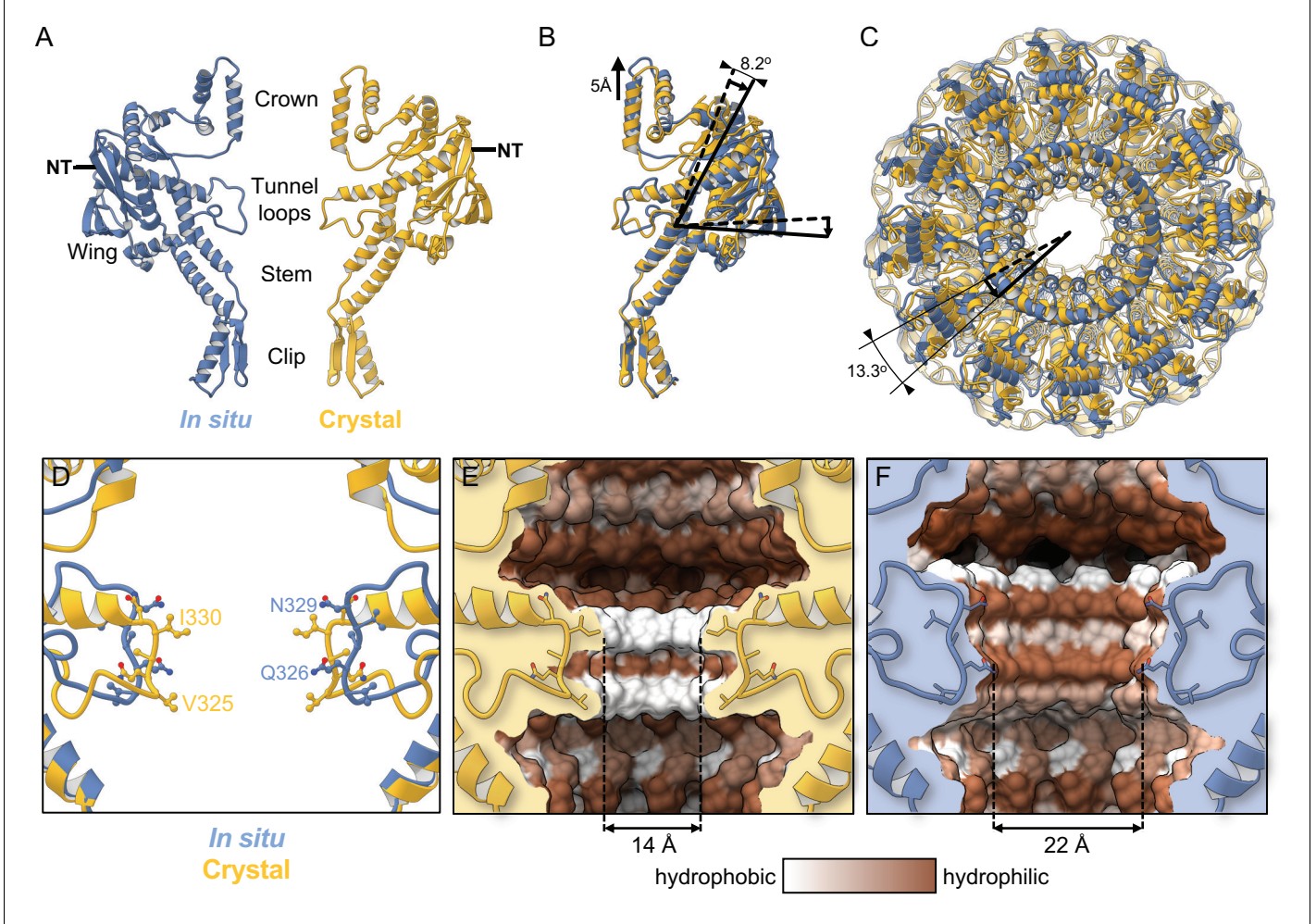

**Figure 2.** Comparison with the crystal structure. (A) Single subunit of the in situ structure is in blue and an apposing chain from the crystal structure is in yellow. (B) Superposition of single subunits, exposing structural differences between the crystal structure and the in situ structure. The curved arrow indicates the pivoting of the Wing domain by ~8° in the in situ structure. (C) The two dodecamers overlaid, viewed from Crown (top domain in A), along the tunnel axis. Dodecamers are superposed based on residues 26–376 (Clip, Stem, and Wing), revealing a ~13° rotation of the Crown domain about the tunnel axis. (D) Overlay of in situ (blue) and crystal structure (yellow), ribbon diagram, with side-chains shown. (E) Van der Waals surface of the crystal structure (PDB 6IBG) showing tunnel loop-constricted region, with tunnel colouring by the hydrophobicity on the Kyte-Doolittle scale where white is hydrophobic and brown is hydrophilic, and same for (F) but for the in situ structure (PDB 6QJT). Diameters of most constricted part of tunnels measured from Van der Waals surfaces are shown.

reveals the details of the portal-capsid interactions at this symmetry-mismatched interface. In the asymmetric reconstruction of the procapsid, the portal protein appears 12-fold symmetric (*Bayfield et al., 2019*). Residues 185–189 (β-hairpin loop) of the portal Wing are involved in interactions with the capsid (*Figure 3B*). These loops may pivot downwards to make closer contact with the capsid inner wall, in select chains (*Figure 3B,C*). Such adjustments in specific subunits of the portal protein would not be resolved in a symmetrically averaged structure; however, the bridging regions observed between the capsid and the portal in the asymmetric procapsid reconstruction suggests local deviations from C12 symmetry are possible. The portal Wing (β-hairpin loop) is in close proximity to residues 24–30 and 337–340 of the major capsid protein (*Figure 3B*). Interactions of the portal Clip likely involve portal protein residues 263–275 (α-helix and adjacent loop within the Clip domain) interacting with the major capsid protein residues 119–127 and 357–358 (*Figure 3D*). In common with φ29 (*Simpson et al., 2000*), portal-capsid interactions are mediated by residues of both polar and hydrophobic characters. The portal–capsid symmetry mismatch means that only select portal

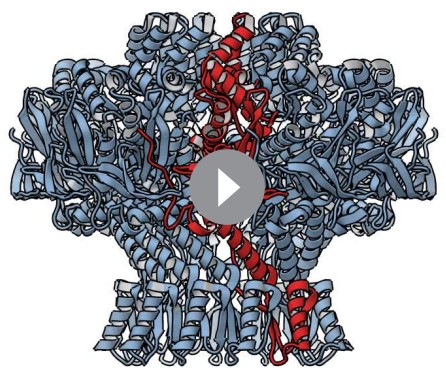

**Video 2.** Morph between the in situ structure (first) and crystal structure (second). Ribbon diagram, first viewed perpendicular to the tunnel axis, then viewed along the tunnel axis, then rotated back to initial view with two apposing chains displayed.

https://elifesciences.org/articles/55517#video2

chains make contact with the capsid: these are chains A-C-(E/F)-H-J at the Wing (*Figure 3E*), and chains C-E-(G/H)-J-L at the Clip (*Figure 3F*).

## Discussion

### Procapsid assembly primes the portal for packaging

The in situ structure of the portal protein differs from the crystal structure globally, in changes in domain positions, and locally, in conformational changes such as the inversion of the tunnel loop. Structural data indicate how the changes on these two levels are linked:

1. Assembly of capsid proteins around the portal stabilises a ~8° rotational adjustment in Wing domains (*Figure 2B*).
2. As the Wing domain pivots, the C-terminal helix of the Crown domain that interacts with the Wing, slides, facilitating a ~5 Å shift of the Crown towards the capsid centre (*Figure 2B*).
3. Movement of the Crown creates space between the Wing and Crown, which allows remodelling of tunnel loops, facilitated by an unfolding of a 5-residue segment of the long helix (residues 331–335, *Figure 2D*) within the Wing domain.
4. The loop remodelling 'switches' the properties of the tunnel surface from hydrophobic to hydrophilic, causing the tunnel to 'open' at its most constricted part (*Figure 2E,F*)
5. Reversal of the Crown and tunnel movements (steps 4 to 1 above) would cause the tunnel to revert to the 'closed' state, as shown schematically on *Figure 4*.

It is reasonable to assume that the two conformational states observed in structural studies, reflect energetically preferred states of the portal protein that are utilised during DNA translocation. The switch between the open and closed states, resulting in alteration of surface properties of the internal tunnel may therefore have a role in the packaging mechanism. The observed conformational differences between the two portal protein states are consistent with the normal mode analysis (*Bayfield et al., 2019*), suggesting that a dynamic equilibrium exists between these two states. Analogous conformational changes in a central tunnel, involving a hydrophobic–hydrophilic 'switch' in surface properties, have been proposed to play a key mechanistic role in other systems, for example the GroEL molecular chaperone, where ATP-induced changes facilitate protein refolding (*Mayhew et al., 1996*; *Weissman et al., 1996*; *Xu et al., 1997*).

It is important to consider how the two portal states are related and how they may participate in the DNA translocation mechanism. Whereas the ~22 Å-wide hydrophilic tunnel observed in situ would allow the passage of B-form and potentially even the wider A-form DNA (*Harvey, 2015*; *Ray et al., 2010*) into the capsid, the more restrictive tunnel diameter of ~14 Å observed in the crystal structure requires the tunnel loops to protrude towards the DNA grooves, involving changes in the tunnel loop conformations (*Lebedev et al., 2007*).

### Mechanism preventing DNA slippage during translocation

Based on structural observations, we propose the following mechanism (*Figure 4*). At the start of a packaging cycle, the Crown is protruding towards capsid and hence the tunnel is open for DNA translocation (*Figure 4*) and its internal surface is hydrophilic (*Figure 2F*). As shown for the φ29 system, DNA is expected to be translocated in bursts followed by dwell intervals, serving to reset the motor (*Chistol et al., 2012*; *Moffitt et al., 2009*). When the packaging driving force is removed, as when the motor is resetting to bind ATP (*Feiss and Rao, 2012*), or when the motor fully detaches in

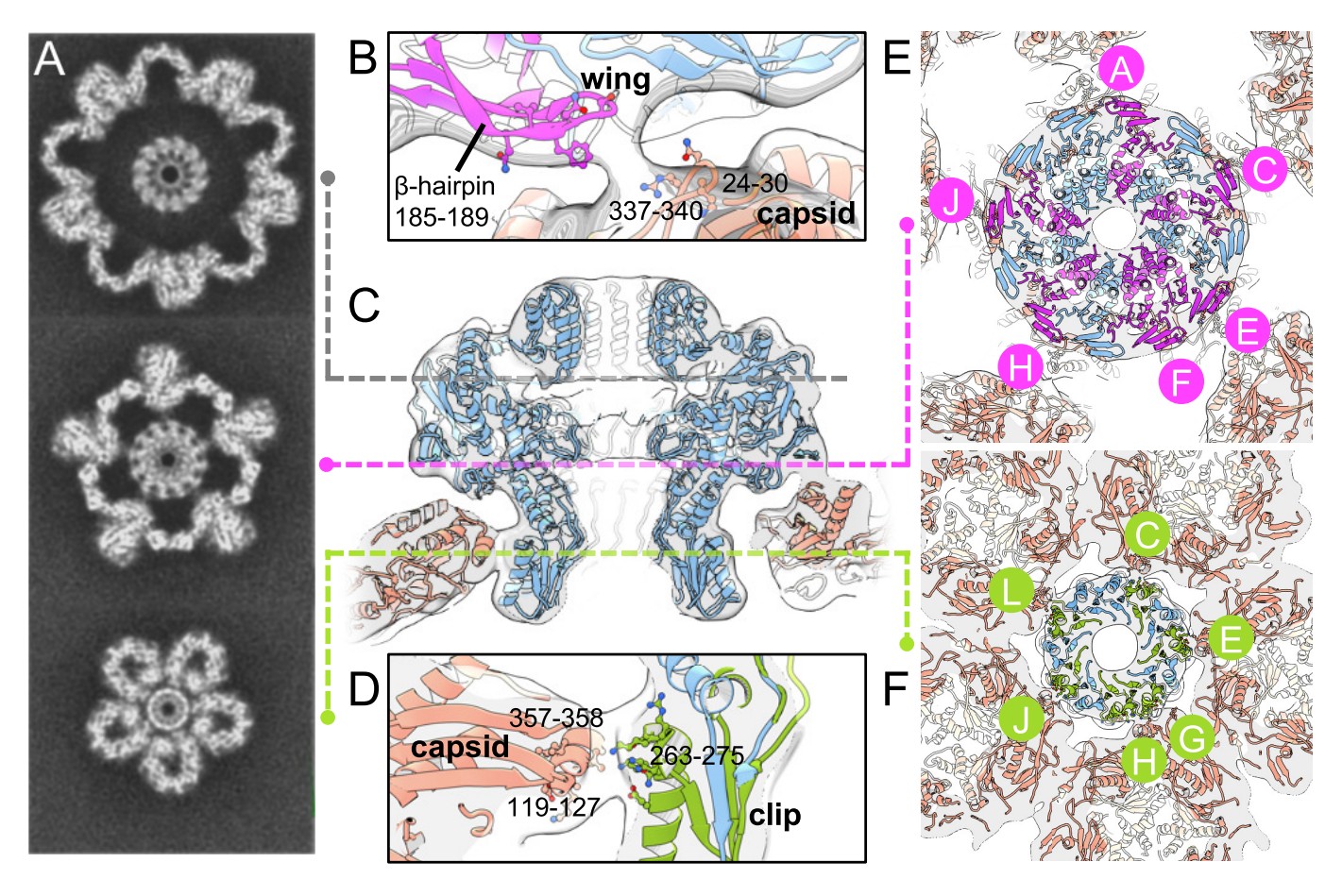

**Figure 3.** Portal–capsid interactions. (**A**) Sections through the capsid reconstruction perpendicular to the portal tunnel axis, at three different heights as denoted on (**C**) by dashed lines. (**B**) Interactions between the portal Wing and capsid. Portal protein subunit making interactions with the capsid is in pink. Portal subunits not making interactions are in blue. (**C**) Ribbon diagram of the in situ portal protein fitted into the procapsid map. (**D**) Interactions between the portal Clip and capsid. Portal protein subunit making closest interactions with the capsid is in green. (**E**) Subunits of the portal protein interacting with the capsid by their Wing regions are in magenta, labelled clockwise. (**F**) Subunits of the portal protein interacting via their Clip are in green. View is from the center of the portal with chains labelled as in (**E**).

preparation for tail docking (*Cuervo et al., 2019*), the risk of the genome leaking from the capsid increases. This risk is highest when the pressure inside the head is at its greatest, when the head becomes fuller. In this instance, the portal tunnel can act to negate this risk, constricting to prevent DNA from slipping out (*Figure 4*). In this scenario, the tunnel loops engage with DNA to prevent its slippage, in a manner analogous to a ratchet. This would be caused by the downward movement of the Crown, pushing on the tunnel loops. Such a mechanism is consistent with variation in the length of packaging dwell periods, which become longer as the capsid fills, as observed for the φ29 system (*Chistol et al., 2012*; *Liu et al., 2014*; *Moffitt et al., 2009*), and with the arresting DNA slippage, as observed in 'single molecule' experiments for T4 (*Ordyan et al., 2018*).

As a result of this synergy in the movement of portal Crown domain and tunnel loops, the closed state could be induced more readily by a higher internal pressure pushing on the Crown domain, which builds as the capsid fills with DNA, or by the occasional slippage of DNA which could interact transiently with the Crown ('snagging'). The role of the tunnel loops in engaging with the DNA, particularly during the late stages of packaging, is supported by the observations that tunnel loop deletions allow DNA to escape from the capsid in phages φ29 and T4 (*Grimes et al., 2011*; *Padilla-Sanchez et al., 2014*). Overall, this describes a packaging mechanism that is naturally safeguarded against genome loss by the portal protein. When fully packaged, DNA can be held in place by the

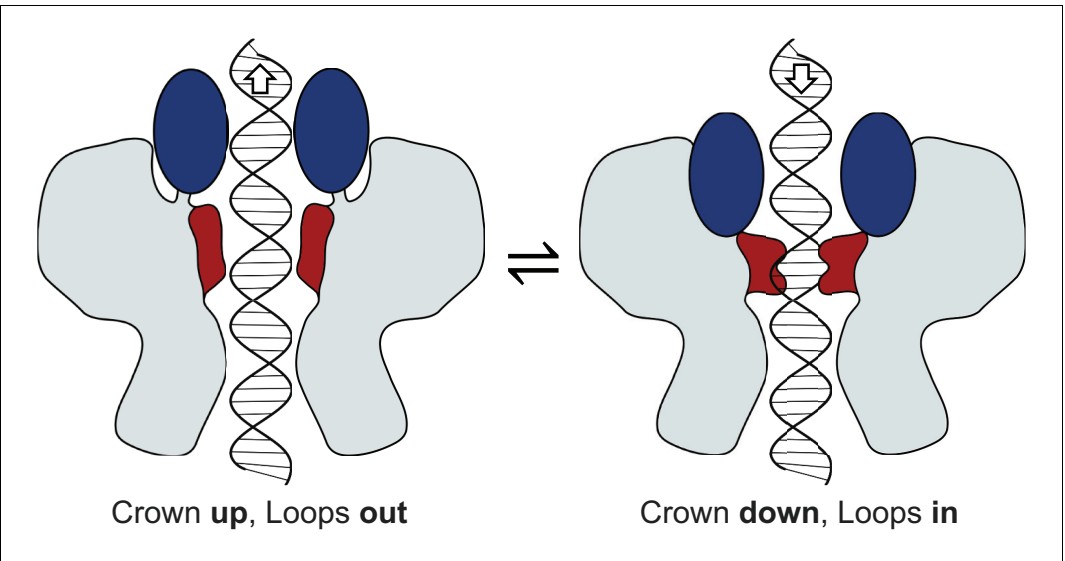

**Figure 4.** Mechanism of portal tunnel closure. Left - the open state where the Crown (blue) is elevated, facilitating partial retreat of the tunnel loops (terracotta) toward the crown, widening the tunnel. Right – the closed state where the Crown is depressed into the body of the portal protein, facilitating closure of the tunnel where tunnel loops adopt a conformation that extends into the tunnel.

constricted tunnel (*Liu et al., 2019*; *McElwee et al., 2018*) and by tail factors that completely block DNA exit (*Cuervo et al., 2019*). During infection and DNA ejection, bacterial cell surface binding is likely able to influence the conformation of the phage tail and consequently the portal protein, inducing a more open conformation needed for DNA escape. Due to the nature of portal-capsid interactions and the attendant symmetry mismatch, discussed below, the portal ratcheting mechanism could be active, regardless of the capsid expansion state.

## The portal's high order of symmetry reconciles a symmetry mismatch

The C12-symmetric portal is accommodated in a C5-symmetric penton cavity at one capsid vertex, despite the attendant symmetry mismatch. Comparison of the C12 portal reconstruction presented here with the asymmetric procapsid reconstruction (*Bayfield et al., 2019*), reveals how this is achieved. The ~8° rotational adjustment of the Wing position, bringing it closer to the capsid wall, may assist in forming close portal–capsid contacts, whereas the portal's Clip external diameter is already well matched to the aperture of the capsid's penton hole (i.e. the space vacated if one complete penton is removed), so that close interactions can be made. However, the symmetry mismatch creates a problem in that the same pairs of interacting residues at the portal–capsid interface are not consistently aligned around all subunits, and could be offset by as much as ~2 nm in the case of P23-45. This misalignment occurs both at the portal Wing and at the Clip, where the portal and capsid make contact in the asymmetric capsid reconstruction. The sparsity of connected portal–capsid regions indicates that the total surface area of interaction is small, and the residues involved in such interactions are hence also restricted in number and positioning.

The high order of symmetry of the portal helps to mitigate these problems. Its 12-fold symmetry is advantageous in that regions of the portal which can interact with the capsid are repeated at a correspondingly high frequency, which reduces the distance between the mismatched interacting residues. The remaining distance can easily be closed by pivoting of flexible loops towards the capsid, such as at the β-hairpin loops of the portal Wing (residues 185–189). These loops are in equivalent positions in φ29 (*Xu et al., 2019*). As a result, only minimal, localised deviations from ideal C12 symmetry are needed to make interactions with the capsid. The portal can therefore utilise the same few residues to interact around its circumference, which contrasts with the situation that would exist if the portal possessed C6, C3, or other lower symmetries matching that of tail components. The symmetry mismatch of the interaction is a general feature amongst all tailed bacteriophages and related viruses, including herpesviruses (*Liu et al., 2019*; *McElwee et al., 2018*). In the case of

bacteriophage φ29 prohead (*Mao et al., 2016*; *Xu et al., 2019*), one of the structural roles of the pRNA appears to be equivalent to that of the capsid protein P-domain, in interacting with the outside of the portal Clip, with the φ29 capsid protein P-domain, instead making contact with an N-terminal segment of the portal protein.

At the interface between the portal and capsid vertex, with respective C12 and C5 symmetries, interactions will repeat with a periodicity of 360°/60 = 6°, as similarly suggested by Hendrix prior to the determination of portal structures (*Hendrix, 1978*). Rotation of the portal with respect to the capsid by only 6° would therefore create an equivalent global register, with rotations less than 6° generating non-superposable registers of the whole capsid particle (*Video 3*). Different portal–capsid registers will have different energies of interaction, and hence equivalent angular registers are expected to be energetically equivalent. A comparable symmetry mismatch is observed between the portal and internal core of bacteriophage T7 (C12 *versus* C8) (*Cerritelli et al., 2003*), where the mismatched interactions may facilitate the detachment of core proteins. As neither detachment of the portal nor its rotation with respect to the capsid (*Baumann et al., 2006*), appear to play a role in capsid maturation, the effect of symmetry mismatch in the capsid vertex is to permit flexibility at the portal–capsid interface, allowing the portal and capsid to undergo independent conformational changes, whilst ensuring stable interaction of the portal protein with the tail.

## Conclusions

Accommodation of the portal protein dodecamer in the procapsid involves conformational adjustments. Interaction between the portal and the capsid shell alters the relative positions of domains, in particular the Wing and Crown, and causes remodelling in the tunnel loops that define the most constricted part of the axial tunnel. The unique conformation of the portal in situ demonstrates that the capsid plays a role determining portal conformation, allowing DNA to pass through the tunnel, whilst the portal has the ability to modulate packaging activity and slippage by switching its tunnel properties so that it can engage and disengage with DNA. Whilst portal proteins across other double-stranded DNA viruses (with a terminase motor) may deviate from the classical domain arrangement observed in P23-45, all such viruses face the same basic challenge of safeguarding against genome loss. With regards to portal-capsid interactions, the adoption of 12-fold symmetry by the portal, rather than a symmetry matching that of the capsid vertex, is likely a consequence of the independent evolution of head and tail assemblies, which has selected the matching of symmetries between the portal (12-fold) and the tail (6-fold). This study posits that the problem of mismatched portal–capsid interactions is resolved by the large number of subunits constituting the portal protein, which minimises distances between interacting regions across a spacious interface.

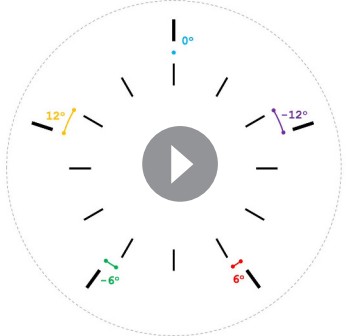

**Video 3.** Portal–capsid registers. One-degree step change in portal register (inner 12-fold circle) with respect to capsid vertex (outer 5-fold circle), beginning with '0'. Portal register '6°' is superposable on register '0°' by 144° rotation of the whole capsid (i.e. rotating both inner and outer circles together).
https://elifesciences.org/articles/55517#video3

## Materials and methods

### Cryo-EM data processing and model building

From 38,044 extracted particles used in the reconstruction of the unexpanded icosahedral capsid (EMD-4447) (*Bayfield et al., 2019*), subparticles centred on each vertex were extracted from each capsid particle, and aligned on the z-axis (*Ilca et al., 2015*). After 3D classifications without imposing symmetry or changing orientations in RELION (*Scheres, 2012*), a class containing 10,025 particles and exhibiting clear portal features was selected for subsequent 3D refinement in RELION, with C12 symmetrical averaging. The atomic model was built using the crystal structure PDB 6IBG as a starting model, with modification to domain positions and to individual amino acids, including side-chain conformations, introduced in Coot (*Emsley and Cowtan,*

*2004*). Cycles of model rebuilding were followed by real-space refinement in PHENIX (*Adams et al.,* *2010*). Resolution was assessed using the FSC 0.143 criterion. Refinement and model statistics are presented in *Supplementary file 1*. Rendering of figures and structure analyses was performed in UCSF Chimera (*Pettersen et al., 2004*) and ChimeraX (*Goddard et al., 2018*).

### Liquid chromatography–mass spectrometry

Capsids of P23-45 in the unexpanded state were purified as previously described (*Bayfield et al.,* *2019*), and digested with enzyme Glu-C, followed by liquid chromatography tandem mass spectrometry. A 20 μl aliquot (20 μg protein) was reduced with DDT and alkylated with iodoacetamide. Digestion was performed for ~18 hr at 37 °C using sequencing grade Glu-C (Promega). Peptides were analysed by nanoHPLC-MS/MS over a 65-min acquisition with elution from a 50-cm C18 Pep-Map column onto an Orbitrap Fusion Tribrid mass spectrometer via an EasyNano ionisation source. LC-MS/MS chromatograms were analysed using PEAKS-Studio X (*Tran et al., 2019*). Peaks were picked and searched against the combined *Thermus thermophilus* and Thermus phage P23-45 proteomes. Database searching required Glu-C specificity with one site of non-specificity per peptide identity allowed. Expected cleavage is C-terminal to Glu, a lower rate of cleavage C-terminal to Asp is also known to occur. Peptide matches were filtered to achieve a false discovery rate of <1%.

## Acknowledgements

The authors thank Emma Hesketh and Rebecca Thompson at the Astbury Centre, University of Leeds, for assistance with cryo-EM data collection and helpful discussion; Adam Dowle at the Biology Technology Facility, University of York, for assistance with mass spectrometry. This work was supported by Wellcome Trust–National Institutes of Health Studentship 103460 (to OWB), by the Intramural Research Program of National Institute of Arthritis and Musculoskeletal and Skin Diseases (to ACS) and by Wellcome Trust fellowship 206377 (to AAA).

## Additional information

### Funding

| Funder | Grant reference number | Author |
| --- | --- | --- |
| Wellcome | 103460 | Oliver W Bayfield |
| Wellcome | 206377 | Alfred A Antson |
| National Institute of Arthritis and Musculoskeletal and Skin Diseases | Intramural Research Program | Alasdair C Steven |
| National Institutes of Health | 103460 | Oliver W Bayfield |

The funders had no role in study design, data collection and interpretation, or the decision to submit the work for publication.

### Author contributions

Oliver W Bayfield, Conceptualization, Data curation, Formal analysis, Validation, Investigation, Visualization, Methodology, Writing - original draft, Writing - review and editing; Alasdair C Steven, Conceptualization, Funding acquisition, Investigation, Writing - original draft, Writing - review and editing; Alfred A Antson, Conceptualization, Resources, Data curation, Funding acquisition, Investigation, Writing - original draft

### Author ORCIDs

Oliver W Bayfield https://orcid.org/0000-0003-1421-7780
Alfred A Antson https://orcid.org/0000-0002-4533-3816

### Decision letter and Author response

Decision letter https://doi.org/10.7554/eLife.55517.sa1

Author response https://doi.org/10.7554/eLife.55517.sa2

---

## Additional files

### Supplementary files
• Supplementary file 1. Cryo-EM data collection and refinement statistics.

• Transparent reporting form

### Data availability
Cryo-EM reconstruction (EMD-4567) and atomic coordinates (PDB 6QJT) have been deposited with the wwPDB (http://www.wwpdb.org/).

The following datasets were generated:

| Author(s) | Year | Dataset title | Dataset URL | Database and Identifier |
|---|---|---|---|---|
| Bayfield OW, Antson AA | 2020 | Cryo-EM reconstruction | https://www.emdatare-source.org/EMD-4567 | EMDataResource, EMD-4567 |
| Bayfield OW, Antson AA | 2020 | Atomic coordinates | http://www.rcsb.org/structure/6QJT | RCSB Protein Data Bank, 6QJT |

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
