## [Decision Letter]

Thank you for submitting your article "Cryo-EM structure in situ reveals a molecular switch that safeguards virus against genome loss" for consideration by *eLife*. Your article has been reviewed by four peer reviewers, including Edward H Egelman as the Reviewing Editor and Reviewer #1, and the evaluation has been overseen by John Kuriyan as the Senior Editor. The following individual involved in review of your submission has agreed to reveal their identity: Paul J. Jardine (Reviewer #2).

The reviewers have discussed the reviews with one another and the Reviewing Editor has drafted this decision to help you prepare a revised submission.

Summary:

The authors report the near-atomic resolution structure of the P23-45 phage procapsid portal complex and focus on changes in the portal in situ compared to the isolated portal, and describe the interactions between the portal oligomer and the capsid vertex.

The results are thoughtfully interpreted, using careful language, and related well to the existing literature. In particular, the report of a rearrangement of the tunnel loops and the consequences of the 5-12 symmetry mismatch between portal and head shell common to all tailed phages suggest a mechanistic role of the former in DNA retention during and after packaging, and an explanation of tolerance for the latter. Overall, this manuscript will add nicely to the field when published, providing new interpretations of the role and organisation of the components discussed and presenting new, testable ideas of their function during assembly and maturation. Although a phage-focused work, the universality of these systems and the applicability of these observations to other high- and mixed-symmetry assemblages merit publication at this level.

Essential revisions:

1) Subsection “The portal’s high order of symmetry reconciles a symmetry mismatch”, first paragraph. The few references to an asymmetric reconstruction are very confusing. The asymmetric reconstruction appears to refer to something in the Bayfield et al., 2019 paper that was never really discussed in that paper. It is shown in Figure 5B in the PNAS paper, and remarkably not discussed in the text of that paper. So the C12 reconstruction in the present manuscript is being compared with a totally asymmetric reconstruction in the PNAS paper, and this needs clarification.

2) The authors should point out that the portal crystal structure used here, PDB 6IBG, is not that of phage P23-45 but instead of the closely-related phage G20c (see Bayfield et al., 2019). While this may have little consequence on the structural interpretation since the two portal proteins are 99.3% identical in sequence, it is not explained that different phage portals are being compared. It may also be worth pointing out on the structure where the 0.7% amino acids are located that are different between the P23-45 cryo-EM structure and the G20c crystal structure, and what the changes might indicate.

3) There is also a lack of clarity on the implications of the portal state. The new structure is the in situ procapsid portal, but the crystal structure is not only from another phage but it is assembly-naïve – i.e., it is purified and self-assembled (as far as can be understood from Bayfield et al., 2019). The presence of a constriction implies that it represents a closed state, which presumably resembles that involved in packaging to prevent leakage of DNA. However, the dynamics presented at the start of the Discussion are that capsid assembly switches the free closed portal into the open form due to capsid-portal interactions. The closure of the portal against the internal pressure of the packaged DNA during dwell times of the packaging motor is supposed to be induced by the DNA itself, acting much as a ratchet (an analogy not used here, perhaps out of fashion?). However, what of further changes in capsid-portal interactions following capsid expansion, and if the portal is then closed against the high DNA pressure in the mature capsid, what opens it again (still in the mature capsid) to release the DNA? This is beyond the scope of the experimental results, but given the previous work by the same group that includes lower resolution asymmetric cryo-EM structures of procapsid and expanded capsids, it seems a natural area for informed speculation.

---

## [Author Response]

Essential revisions:1) Subsection “The portal’s high order of symmetry reconciles a symmetry mismatch”, first paragraph. The few references to an asymmetric reconstruction are very confusing. The asymmetric reconstruction appears to refer to something in the Bayfield et al., 2019 paper that was never really discussed in that paper. It is shown in Figure 5B in the PNAS paper, and remarkably not discussed in the text of that paper. So the C12 reconstruction in the present manuscript is being compared with a totally asymmetric reconstruction in the PNAS paper, and this needs clarification.

We fully agree and have clarified this by adding sentences in the Discussion and Results sections, as follows:

(Results)

“Fitting of the C12 portal preconstruction, presented here, into the asymmetric procapsid reconstruction (Bayfield et al., 2019), reveals the details of the portal-capsid interactions at this symmetry-mismatched interface.”

(Discussion)

“Comparison of the C12 portal reconstruction presented here with the asymmetric procapsid reconstruction (Bayfield et al., 2019),…”

2) The authors should point out that the portal crystal structure used here, PDB 6IBG, is not that of phage P23-45 but instead of the closely-related phage G20c (see Bayfield et al., 2019). While this may have little consequence on the structural interpretation since the two portal proteins are 99.3% identical in sequence, it is not explained that different phage portals are being compared. It may also be worth pointing out on the structure where the 0.7% amino acids are located that are different between the P23-45 cryo-EM structure and the G20c crystal structure, and what the changes might indicate.

This is a valuable point. The use of the G20c structure for this comparison has now been clarified, with the 3 residues that differ between the two structures listed, and the location of residues described in the revised version, which contains the following sentences in the first paragraph of the Results:

1) “Comparison with the crystal structure of the portal protein from the closely related phage G20c (PDB 6IBG, 99.3% sequence identity)….”

2) “Although the two portal proteins compared are from different phage, they have closely related sequences. The most conservative substitution, I328V, is located in the tunnel, and two additional conservative substitutions, S189N and S367G, are located at the outer surface of the wing in residues with solvent-exposed side chains.”

3) There is also a lack of clarity on the implications of the portal state. The new structure is the in situ procapsid portal, but the crystal structure is not only from another phage but it is assembly-naïve – i.e., it is purified and self-assembled (as far as can be understood from Bayfield et al., 2019). The presence of a constriction implies that it represents a closed state, which presumably resembles that involved in packaging to prevent leakage of DNA. However, the dynamics presented at the start of the Discussion are that capsid assembly switches the free closed portal into the open form due to capsid-portal interactions. The closure of the portal against the internal pressure of the packaged DNA during dwell times of the packaging motor is supposed to be induced by the DNA itself, acting much as a ratchet (an analogy not used here, perhaps out of fashion?). However, what of further changes in capsid-portal interactions following capsid expansion, and if the portal is then closed against the high DNA pressure in the mature capsid, what opens it again (still in the mature capsid) to release the DNA? This is beyond the scope of the experimental results, but given the previous work by the same group that includes lower resolution asymmetric cryo-EM structures of procapsid and expanded capsids, it seems a natural area for informed speculation.

We agree that the analogy of a ratchet is apt and, in fact, was included in earlier drafts. This has now been reintroduced in the Discussion to aid in describing the proposed mechanism.

The reviewer raises important points concerning (a) the role of the expanded capsid state and its influence on the portal’s conformation, and (b) the DNA ejection mechanism and how a closed state can permit DNA exit during infection.

Concerning point (a), we expect the portal-capsid interactions at the Wing of the portal to be similar in the expanded state, whilst those at the Stem/Clip of the portal should differ in order to allow the tail to bind there. Given that DNA packaging can proceed in both unexpanded and expanded capsid states, we propose the portal’s influence on this process, and its ratcheting effect, to be similar in both states. We (and others) have observed the expanded state to be slightly superior than the procapsid state at protecting DNA in packaging assays. This suggests the expanded state is at least no worse in its ability to package and to prevent slippage. This notion is further supported by the symmetry mismatch and hence the expected lack of preferred portal-capsid orientation, as we allude to in the Discussion. The portal’s ratcheting mechanism can exist, regardless of the expansion state, and the capsid’s motions can be considered separately from those of the portal. This is consistent with such a mechanism existing in the phi29 system, where expansion is not really a feature, but the need to prevent slippage still exists. To address this issue, we have added the following sentence in the Discussion that links the ratcheting mechanism to the expansion state and in turn the portal-capsid interactions:

“Due to the nature of portal-capsid interactions and the attendant symmetry mismatch, discussed below, the portal ratcheting mechanism could be active, regardless of the capsid expansion state.”

Regarding point (b), DNA must indeed be allowed to now slip, fully, from the capsid, switching from a closed state in the mature virion. We expect the infection process, specifically the binding of tail factors to the bacterial cell surface, to bring about changes in the portal *via* its interaction with the tail, that allow the tunnel to recover an open conformation. This state may be observed only transiently, during the process of DNA ejection. A sentence has been added in the Discussion to address this important question.

“During infection and DNA ejection, bacterial cell surface binding is likely able to influence the conformation of the phage tail and consequently the portal protein, inducing a more open conformation needed for DNA escape.”